# Phenotypic and Genomic Analysis of *Enterobacter ludwigii* Strains: Insights into Mechanisms Enhancing Plant Growth Both Under Normal Conditions and in Response to Supplementation with Mineral Fertilizers and Exposure to Stress Factors

**DOI:** 10.3390/plants13243551

**Published:** 2024-12-19

**Authors:** Ekaterina Alexeevna Sokolova, Olga Viktorovna Mishukova, Inna Viktorovna Hlistun, Irina Nikolaevna Tromenschleger, Evgeniya Vladimirovna Chumanova, Elena Nikolaevna Voronina

**Affiliations:** 1Institute of Chemical Biology and Fundamental Medicine, Siberian Branch of the Russian Academy of Sciences, 630090 Novosibirsk, Russia; sokolova_ea@niboch.nsc.ru (E.A.S.);; 2Department of Natural Sciences, Novosibirsk State University, 630090 Novosibirsk, Russia

**Keywords:** *Enterobacter ludwigii*, plant growth-promoting bacteria, *Triticum aestivum*, mineral fertilizer, drought, salinization, reactive oxygen species

## Abstract

In this research study, we investigated four strains of *Enterobacter ludwigii* that showed promising properties for plant growth. These strains were tested for their ability to mobilize phosphorus and produce ammonium, siderophores, and phytohormones. The strains exhibited different values of PGP traits; however, the analysis of the complete genomes failed to reveal any significant differences in known genes associated with the expression of beneficial plant traits. One of the strains, GMG_278, demonstrated the best potential for promoting wheat growth in pot experiments. All morphological parameters of wheat were improved, both when GMG_278 was applied alone and when combined with mineral fertilizer. The combined effect we observed may suggest various mechanisms through which these treatments influence plants. The amount of pigments and proline suggests that bacterial introduction operates through pathways likely related to stress resilience. A study on the genetic mechanisms behind plant resilience to stress has revealed a significant upregulation of genes related to reactive oxygen species (ROS) defense after bacterial exposure. It is important to note that, in the initial experiments, the strain showed a significant production of salicylic acid, which is a potent inducer of oxidative stress. In addition, the synthesis of some phytohormones has been restructured, which may affect root growth and the architecture of root hairs. When combined with additional mineral fertilizers, these changes result in a significant increase in plant biomass.

## 1. Introduction

Climate factors, such as extreme temperature conditions (heat, cold, and frost), drought caused by a lack of precipitation and drying winds, and soil contamination with high concentrations of salt, are the main abiotic stressors that limit plant growth, development, and, consequently, agronomic productivity [1]. These factors play an important role in determining the geographic distribution of plant species. The reactions to environmental stimuli occur at all levels of biological organization. At the cellular level, these responses include modifications in the structure of the membrane system and the architecture of the cell wall, as well as alterations in the cell cycle and division processes. Furthermore, plants exhibit a diverse array of metabolic changes in response to stressors [2]. These processes involve the synthesis of compatible solutes, such as proline, raffinose, and glycine betaine. These compounds play a crucial role in stabilizing proteins and cellular structures, as well as maintaining cell turgor through effective osmotic regulation. Additionally, these mechanisms include modifications in redox metabolism aimed at reducing excessive levels of reactive oxygen species (ROS), thereby restoring cellular redox homeostasis [3].

One of the strategies for increasing plant stress resistance involves the use of beneficial bacteria. Microbial biostimulants can have a positive impact not only on crop yield but also on the quality of the product by increasing the levels of essential nutrients and bioactive compounds beneficial to human health [4]. Numerous studies have demonstrated that the use of plant growth-promoting bacteria (PGPB) represents a relatively simple and cost-effective alternative approach that can positively impact growth indices, biomass production, and chlorophyll content in plants, even in stressful conditions. In particular, various bacterial strains, including *Rhizobium, Pseudomonas, Bacillus*, and *Enterobacter,* have demonstrated their ability to enhance crop growth and productivity under ordinary and stressful conditions [5]. PGPB recognize and further reprograms gene expression patterns in plants to help them overcome both biotic and abiotic stresses through the activation of induced systemic resistance.

*Enterobacter* are facultatively anaerobic, Gram-negative, motile bacteria that possess peritrichous flagella. They have been shown to exhibit a number of characteristics that contribute to plant growth, such as the ability to fix nitrogen, solubilize phosphorus in the soil, and produce various secondary metabolites, including antibiotics, siderophores, extracellular polysaccharides, and enzymes such as chitinases and ACC deaminases. Additionally, these bacteria can improve soil porosity and structure. [3,6,7]. A previous study [8] showed that *E. cloacae* actively made a settlement on maize roots and induced salicylic acid-dependent systemic acquired resistance; as a result, pre-colonized plants gained the ability to resist *Fusarium* infection. *Enterobacter ludwigii* strain AFFR02 has been described as having a high ability to solubilize phosphates and growing on high-PEG-containing media. In studies on alfalfa inoculated with this strain under drought stress, recovery of growth and mass indices was revealed on shoots and roots [9]. *Enterobacter ludwigii* SA K5 was identified to have a high production of indole-3-acetic acid (IAA), siderophores, and phosphate solubilizers. This strain significantly mitigated the adverse effects of salt stress and enhanced the growth of rice plants under salt-stress conditions compared to non-inoculated plants. Simultaneously, it was observed that in inoculated plants, the levels of abscisic acid (ABA) decreased significantly while the levels of glutathione increased [10]. Inoculation with *Enterobacter asburiae* NC16 in maize (*Zea mays*) plants reduced the rate of water loss through transpiration and the expression of several genes involved in iron (Fe) uptake, including ZmFer, ZmYS1, and ZmZIP. This led to a reduction in cadmium toxicity, at least partially, by inhibiting Fe uptake-related pathways [11]. The above examples demonstrate the ability of bacteria to impact plant adaptation to stressful conditions through different mechanisms.

Combinations of mineral and organic fertilizers are also seen as a promising area of research. The addition of mineral fertilizers to selected microorganisms or other bioactive components can facilitate the conversion of chemicals into forms that are more readily available to plants while also providing a transport function to ensure that essential chemical elements are absorbed by plant roots and do not pollute the environment. In particular, the research conducted by Biswas et al. demonstrated that the incorporation of low-grade rock phosphate (LGRP) inoculated with phosphate-solubilizing bacteria (PSB) can significantly enhance plant phosphorus (P) uptake compared with uninoculated treatments. The authors consider that the combination of PSB and LGRPs can reduce the use of standard phosphate fertilizers by 50% without negatively affecting yield [12]. In addition, an increase in the yield of barley (7–17%) was shown to be reached using the same rate of mineral fertilizer by enriching it with bacterial inoculants [13].

In our work, we selected several strains of *Enterobacter ludwigii* collected during the Atlas of Soil Microorganisms project, in the framework of which schoolchildren collected soil samples from different regions of Russia. The selection was based on their ability to grow on a nitrogen-free medium. The aim of the study was to identify various features of these strains, which may be associated with properties beneficial to plants, both at the phenotypic and genotypic levels. In addition, we studied the effect of one of the strains on plant growth under stressful conditions.

## 2. Results

### 2.1. Selection and Phenotypic Screening of Enterobacter ludwigii Strains

The strains of *Enterobacter ludwigii* were selected during the “Atlas of Soil Microorganisms” project, in the framework of which schoolchildren collected soil samples from different regions of Russia; the strains were grown on a nitrogen-free medium and transferred to the Institute of Chemical Biology and Fundamental Medicine of the Siberian Branch of the Russian Academy of Sciences. All the obtained strains were purified and tested for growth activity in a nitrogen-free medium, a medium with insoluble phosphate medium, and a CAS agar medium.

Strains were identified using 16S rDNA sequencing. Among the examined strains of the genus *Enterobacter*, representatives of a certain species, namely *Enterobacter ludwigii*, predominated. Further investigation was conducted using representatives of this species; only strains with the best growth rates on nitrogen-free medium, medium with insoluble phosphate, and CAS agar were selected.

As a result, we selected four strains for the study and carried out a quantitative analysis of various features affecting plant growth for each of them (Table 1). Strain GMG_291 stood out as the top performer in terms of PGP characteristics since it had the highest indices of phosphate solubilization, ammonium production, and the ability to form a biofilm. Strains GMG_336 and GMG_378 were distinguished by high auxin production, while strain GMG_278 showed a high level of production of gibberellin and salicylic acid.

### 2.2. Experiments with Plants

The experiments were carried out under greenhouse conditions; plants were grown in 0.25 L pots. Five wheat seeds were planted in each pot; for each type of treatment and control, three pots were used. The control pots were irrigated with water (Water). For the experimental groups, irrigation was carried out using the following solutions: mineral fertilizer mixture (MF), water with a single addition of bacteria (GMGnumber strain), and mineral fertilizer mixture with a single addition of bacteria (MF_ GMGnumber strain). The results of the experiments are presented in Figure 1 (experimental data are given in Appendix A).

Significant differences in the growth and mass of the above-ground part of plants were observed upon treatment of plants with strains GMG_336 and GMG_278, while changes in the root mass were revealed only towards strain GMG_278. In addition, treating plants with strain, GMG_278 showed a significantly better rate of growth in the case of feeding plants with mineral fertilizers. In general, this result was consistent with the fact that gibberellic acid is the main phytohormone determining plant growth, and mineral feeding helped to realize this potential.

### 2.3. Genome Analysis

Whole genome sequencing followed by de novo genome assembly was performed for the four *Enterobacter ludwigii* strains studied (Table 2).

The species was confirmed after genome-wide sequencing based on the results of BLAST analysis of 16S rRNA and OrtoANI analysis (Figure 2).

The obtained full genome sequences of the four strains were analyzed for the occurrence of genes annotated in the KEGG database. Then, the genes associated with properties beneficial to plants were analyzed. For this purpose, we selected the following features: biofilm production, nitrogen metabolism, siderophores production, as well as production of auxin, gibberellins, and ethylene, phosphorus solubilization, and oxidoreductase production. These data are summarized in Appendix A.

All four bacterial strains showed complete similarity in terms of genes associated with properties beneficial to plants. Overall, these strains were unable to fix atmospheric nitrogen despite their ability to grow on Ashby’s medium; however, they possessed genes for nitrate and nitrite reduction, leading to ammonium production. Additionally, they demonstrated the ability to secrete organic acids that facilitate phosphate solubilization, synthesize siderophores, and produce biofilms. The studied strains contain all the genes necessary for the synthesis of tryptophan, as well as the genes responsible for the conversion of indolepyruvate to indolacetate. These reactions are part of the indolepyruvate pathway of auxin synthesis; however, we have not detected an aminotransferase for the transfer of an amino group from tryptophan to ketoglutarate.

To compare all genes annotated in KeggOrthology, a Venn diagram was constructed (Figure 3). All genes unique to the strains and annotated in the KeggOrthology database are listed in Table 3. It can be noted that strains GMG_278 and GMG_378 were almost identical, except for the only protein with an unknown function. Strain GMG_291 had distinctive genes of proteins of the O-antigen system and DNA repair. Strain GMG_336 had distinctive genes of proteins involved in carbohydrate metabolism and genes of the O-antigen system, as well as several genes of efflux pumps.

In order to assess the genetic differences between the strains, a comparative analysis of open reading frames was performed. Only those open reading frames whose putative protein products are annotated in the KeggOrthology database were included in the comparison. Thus, we can assume the degree of difference between the strains based on the description of the functions of the genes by which the strains differ.

In addition, we identified open reading frames (ORFs) that did not yield homology in the BLAST database. Further, we searched for homology by amino acid motifs in different databases. In this way, we attempted to identify genes that encode proteins with known domains but significantly differ in nucleotide sequence from homologous genes. The results are presented in Table 4, along with the sequences for which homologous amino acid sequences could not be found, and homologs of phage proteins are not given.

In this case, a slightly different situation is observed. Strain GMG_336 contained the fewest unique genes, while strains GMG_278 and GMG_291 showed only minor differences from GMG_336, mainly because of the unique proteins related to antiviral protection. Meanwhile, strain GMG_378 was characterized by significant differences from the other three strains, with a large number of unique genes, many of which could be associated with O-antigen synthesis, as well as distinct genes for antiviral protection. Notably, strains GMG_278, GMG_291, and GMG_336 shared two identical unique genes related to carbohydrate metabolism and peptidoglycan synthesis.

### 2.4. Effect on Plant Growth Under Drought and Salinity Stress Conditions

One of the important aspects of using bacteria is their ability to activate plant resistance against abiotic stress. The only strain GMG_278 was included in this investigation stage since it showed high salicylic acid production in phenotypic tests and was also the most effective in plant tests. Wheat was grown in pots for two weeks under ordinary conditions, as well as under conditions of supplementing with mineral fertilizer. Then, for two weeks, the experimental groups were irrigated with a 1% NaCl solution or a 10% PEG solution. Afterward, morphometric measurements of the plants, as well as determination of chlorophyll A, chlorophyll B, carotenoids, and proline, were carried out (Figure 4 and Appendix A). The results were processed using the principal component analysis (PCA) method and are presented in Figure 5.

The PCA results showed that the principal component 1 mainly reflected the chlorophyll A/B ratio. It is worth noting that the highest ChlA/ChlB ratio was observed in wheat grown under salinity and drought stress. The introduction of bacterial fertilizing had virtually no effect on this indicator, neither in the control nor under stressful conditions. Whereas, in response to mineral fertilizer supplementation, the ChlA/ChlB ratio diminished significantly, but the most significant drop was revealed in the case of a combination of mineral and bacterial fertilizers. This was mainly stipulated by the high level of chlorophyll B in these plants.

Component 2 partially reflected the changes in the ratio of chlorophylls to carotenoids and the proline concentration. The proline concentration increased almost twofold with the introduction of mineral fertilizer both in the control samples and in plants supplemented with bacterium fertilizer. The maximum increase in the proline amount was observed at combined treatment with mineral fertilizer and bacteria. Concurrently, no significant difference was revealed under the stress conditions, except for the case of drought against the background of supplementing with mineral fertilizer.

We revealed that changes in the total amount of chlorophylls and the content of carotenoids showed the same pattern as changes in proline; namely, an increase in their concentrations was observed in plants supplemented with mineral fertilizers. These data attested to an enhancement in pigment synthesis in plants when supplementing them with mineral fertilizers, which was consistent with literary data [19]. Against this background, the ratio of chlorophylls to carotenoids also increased relative to control samples, even under stress conditions.

In addition, the distribution of samples on the PCA plot depended on the morphological characteristics of wheat, including the weight of the above-ground and underground parts, both wet and dry ones, as well as plant growth indicators. Here, we found a positive effect of both mineral fertilization and the introduction of bacteria. Moreover, the manifestations of these effects were almost equivalent. At the same time, the introduction of both of them gave a significant increase in all characteristics, with a maximum value in plants under ordinary conditions and slightly lower values under stressful conditions.

In general, an examination of the heat map (Figure 6) of changes in morphological traits of wheat grown under pure stress or under stress conditions coupled with bacterial feeding showed that the most optimal cultivation option was a combination of mineral salts and bacteria. The grouping of experiments (the tree on the left of the diagram) showed clustering into a control group and a stress factor group, as well as a cluster introducing either mineral fertilizer or bacteria. Concurrently, a cluster combining these two types of plant processing was highlighted separately. One can see that the dry weight of both the above-ground and underground parts of plants was enhanced the most, which may indicate a greater production of plant biomass.

### 2.5. Changes in the Expression of Wheat Genes in Response to the Effects of Bacterium Strain GMG_278 Under Stress Conditions

Plant adaptation to environmental changes requires a new state of cellular homeostasis, which is achieved through a delicate balance between several pathways. Plant cells receive stress signals through various sensors, and the signals are transmitted by various signaling pathways, which involve many second messengers, plant hormones, signal transducers, and transcription regulators [20] (Figure 7). Diagrams of the changes in plant gene expression under stress conditions are shown in Figure 8.

The PCA demonstrated the division of the studied genes into three groups (Figure 9). Principal component 1 (X-axis) reflected the change in gene expression in response to introducing GMG_278 bacteria under any conditions of plant growing. In group 1, which included genes of the reception of stress factors and genes of the first stages of signal transduction, expression remained virtually unchanged. In group 2, which contained genes responsible for the activation of the phytohormonal response under stress conditions, the indicator changed 3–10 times. Group 3, which included the antioxidant defense genes of POD, CAT, and LPX, was characterized by a strong increase in gene expression by tens of times.

Principal component 2 (Y-axis) reflected mainly the response of gene expression to drought (PEG). It should be noted that a zone of positive values included the control samples and plants under drought stress at supplementing with bacteria, while a zone of negative values included plants subjected to drought stress under bacteria-free conditions.

A more detailed examination of gene expression within the groups showed that group 1 was characterized by expression activation only in response to drought effect under conditions of adding bacteria to plants, but the CTR gene expression did not occur in this case. Concurrently, in group 2, equivalent activation was revealed both under drought and under salt stress conditions when supplementing plans with bacteria; in group 3, activation of the POD and CAT genes in response to salinity was significantly higher than in response to drought.

The PCA graph helps to visually assess the dependence of the expression level of plant genes on the growing conditions (PEG, NaCl, Water, MN, and 278), as well as the combination of these conditions. The fact of presence of dependence is the clustering of the feature into groups. The level of assessment of the contribution of growing conditions to the construction of a two-dimensional graph as a percentage on the axis. Three gene clusters are clearly visible on the graph: (1) CTR, MAPK, ABARE, and WKY26; (2) CKX10, AFR2, DREB, and WKY71; (3) LPX, CAT, and POD. The unification into a cluster indicates the similarity of changes in the feature in dependence on the factor, in this case, the expression level on the growing conditions. To identify the conditions that contribute to the development of samples into clusters, the values of PCA 1 and 2 should be analyzed. In this case, the division into clusters in space occurs mainly along the x-axis, so it is necessary to analyze PCA1. The shift in the graph to the right is due to growing conditions with a high positive value (marked in green for convenience). It is clearly seen that all conditions contain a common component—278.

## 3. Discussion

### 3.1. Features of Bacteria Beneficial to Plants in Phenotypic and Genotypic Analyses

In this study, we analyzed the following properties of bacteria of the *Enterobacter ludwigii* species: the production of auxin, gibberellin, proline, salicylic acid, ammonium, siderophores, as well as phosphate-solubilizing activity and the ability to produce biofilms. Among these characteristics were properties aimed at improving the availability of nutrients, including nitrogen and phosphorus, for plants. Another group of features, including the production of auxin, gibberellin, proline, and salicylic acid, was associated with substances that are essential regulators of plant growth and reaction to stress conditions [22,23]. Additionally, the production of salicylic acid and siderophores, as well as the formation of biofilm, are involved in endogenous signaling and protection against pathogens and abiotic stresses.

Among the four strains studied, manifestations of features beneficial to plants varied to a large extent. In particular, strain GMG_278 showed the best ability to protect plants against stress factors and pathogens; at the same time, increased production of gibberellin attested to the possibility of stimulating plant growth, which was later confirmed in experiments on wheat. Concurrently, strain GMG_291 demonstrated the best ability to increase the availability of nutrients, while strain GMG_336 showed good production of auxin and proline.

However, analysis of the complete genomes of these bacteria did not reveal any significant dissimilarity in the known genes involved in the implementation of properties beneficial to plant growth. We attributed this fact to the relatively large number of genes with unannotated functions. A search for genes unique to the studied strains showed that each strain contained genes that had no homologs in the nucleotide sequence database, and some of them had no homologs in the amino acid sequence databases. One can assume that genes, which have not yet been described, or regulatory factors, which control standard genes, make a difference in the phenotypic expression of traits beneficial to plants.

Such discrepancies between phenotypic and genotypic data have been described frequently, especially with regard to nitrogen fixation. In particular, a study of endophytic bacteria in sugar cane found that 23 strains displayed nitrogen-fixing capacity as determined by the acetylene reduction assay; however, only 14 of these strains confirmed the presence of the nifH gene through the use of degenerated primers. Furthermore, the complete genome analysis of *Enterobacter roggenkampii* ED5 revealed that this bacterial species did not possess the complete metabolic pathway for nitrogen fixation [24]. Additionally, the same study described a mismatch in acetylene deaminase activity among the strains, with all strains demonstrating this activity phenotypically, but only 12 strains showing the presence of an acdS gene through genome-wide sequencing. In addition, bacteria that demonstrate the ability to produce IAA from tryptophan do not have a fully developed metabolic pathway for this process, according to whole-genome sequencing data [24]. This has been confirmed in several studies of *Enterobacter* strains, including *Enterobacter* sp. SA187 [25], *Enterobacter* sp. J49 [26], and other bacteria such as *Priestia filamentosa* strain AZC66 [27] and *Bacillus altitudinis* FD48 [28].

Whole genome analysis of *Enterobacter ludwigii* strains was performed on the clinical isolate *Enterobacter ludwigii* type strain EN-119 [29], the metal-resistant strain *Enterobacter ludwigii* NCR3 [30], and *Enterobacter ludwigii* ZCR5 [31], isolated from soil contaminated with hydrocarbons and heavy metals. The latter strain was studied in terms of its effect on plant growth, as well as the presence of genes beneficial for plant growth. Similar to strains investigated in the present study, the genome contained genes associated with phosphate mobilization, siderophore production, and biofilm formation. It should be noted that the strain did not produce siderophores in phenotypic tests. Probably, this fact can be explained by the large variety of siderophores and the inability to determine all sidereophores by a phenotype test. The authors also marked that the analysis of the transcription level of these genes in the rhizosphere and endosphere of *L. perenne* did not show their expression during the phytoremediation experiment. This result allowed the authors to suggest that the activity of the inoculated strains was limited in time or that the stimulation of plant growth by bacteria was caused by the fertilization of plants with dead bacterial biomass. They also assumed that the plant growth stimulating effect could be stipulated by the activity of mechanisms other than those tested in the laboratory. Additionally, it cannot be excluded that some properties involved very different mechanisms, which were not investigated in the gene expression experiment.

### 3.2. The Effects of Mineral Fertilizers and Bacteria Strain Enterobacter ludwigii GMG_278 on the Morphometric and Physiological Features of Wheat

The use of high doses of chemical fertilizers, especially nitrogen fertilizers, for enhancement of plant growth and yield increases the cost of production. In addition, most of the introduced mineral fertilizers are not available to plants and are lost through such processes as leaching, erosion, volatilization, fixation, runoff, denitrification, and precipitation. The availability of minerals can be enhanced by introducing PGPB, which has positive effects on plant growth and the environment [32]. The term “bacteria-impregnated fertilizer” refers to the coating of chemical fertilizers with PGPB to increase the amount of beneficial microflora in the rhizosphere and improve the efficiency of the applied fertilizers [33]. In particular, PGPB *Azospirillum brasilense* in combination with N-fertilizer at 30 or 60 kg/ha enhanced the lettuce height, crown diameter, leaf number, and fresh weight [34]. Nitrogen, phosphorus, and potassium uptake, as well as maize yield, were higher at treatments including mixed bacterial cultures compared with NPK fertilizer alone (nitrogen, phosphorus, and potassium) or NPK + biochar [35]. The relative agronomic efficiency index showed that the combination of biological fertilizer with 50% and 75% NP fertilizer increased cabbage yield by 66% and 48%, respectively, compared with the commercial NP dosage without PGPB [34].

In our experiments, all morphological indices of wheat improved in response to both the application of 50% mineral feeding and supplementation with the GMG_278 strain. The weight gain of the above-ground and underground parts, in both wet and dry states, was approximately 50%, except for the case of the increase in the dry weight of the above-ground part when mineral fertilizer was applied, which resulted in an increase of 107%. In terms of the plant growth indices, the increase was approximately 25% in both cases. The combined application of bacteria and mineral fertilizers resulted in more than a twofold increase in weight for each indicator, with the weight of the above-ground part reaching a 261% increase and growth indices showing an 87% rise. Thus, this demonstrates a cumulative effect of the bacteria and mineral fertilizer on the morphological indicators of wheat, suggesting that these treatments may influence plants through different mechanisms.

The levels of chlorophylls A and B, as well as carotenoids, increased significantly with mineral supplementation, while the addition of bacteria had no significant effect on these indicators. This finding contrasts with some studies. For instance, Mona S. Agha et al. showed that, compared with the control, the highest increase in Chl. a, Chl. b, and carotenoid content was observed in plants inoculated with a combination of *Bradyrhizobium japonicum* + *Enterobacter Delta* PSK [34,36]. In the study by Komal K. Bhise et al., the isolate *Enterobacter cloacae* KBPD increased chlorophyll content in V. radiata L. under salt stress (induced by 50, 100, and 150 mM NaCl) by 19.31, 15.42, and 43.20% respectively compared with non-bacterial seedlings under stress [37]. The authors attributed this effect mainly to improved mineral nutrient availability, driven by both bacterial activity and enhanced root growth. In our study, we assumed that inoculation with the strain *Enterobacter ludwigii* GMG_278 activated wheat growth predominantly through other mechanisms, not related to an increase in nutrient availability.

It is known that the Chl.a/Chl.b ratio increases under stress conditions because of the higher rate of chlorophyll B disintegration and its conversion to chlorophyll A [38]. In our study, when introducing only mineral fertilizer or bacteria, the ChlA/ChlB ratio remained virtually unchanged, while it significantly dropped when supplementing plants with a combination of mineral and bacterial fertilizers. On this basis, we concluded that bacteria somehow affected the retaining of Chl b in plants, accumulated in response to mineral feeding. Meanwhile, the ratio of the sum of chlorophylls a + b to carotenoids had the opposite tendency and increased sharply at combined treatment with mineral fertilizers and bacteria. It is considered that a decrease in the ratio of chlorophylls a + b to carotenoids is an indicator of plant stress since chlorophylls degrade faster than carotenoids [38], and in addition, carotenoids have an antioxidant effect. In our study, the sum of chlorophylls increased with mineral supplementation, and the amount of carotenoids did not change significantly.

Proline is an amino acid that plays a very beneficial role in plants exposed to various stress conditions. In addition to its function as an excellent osmolyte, proline has three other functions under stress, manifesting itself as a metal chelator, an antioxidant defense molecule, and a signal molecule. Stressful environments lead to the overproduction of proline in plants, which increases their stress resistance by maintaining cell turgor or osmotic balance, stabilizing membranes, and preventing electrolyte leakage, as well as by retaining ROS within normal margins and preventing oxidative burst in plants [39]. In our study, introducing mineral fertilizers and bacteria slightly increased the amount of proline, likely due to a common enhancement of cellular metabolic activity; however, introducing both bacteria and mineral fertilizers caused a significant increase in the proline concentration, indicating the activation of plant stress resistance mechanisms. In general, such activation in response to bacterial inoculation has already been described as one of the factors in the emergence of plant resistance against injuring factors [36,40]. In addition, *Enterobacter* was shown to significantly reduce fungal invasion because of its ability to stimulate the production of osmoprotectants such as proline, increase the activity of antioxidant enzymes, and reduce the cell membrane permeability in maize plants at different growth stages [8].

Thus, we established that inoculation with the *Enterobacter ludwigii* GMG_278 strain had a beneficial effect on the morphometric indices of wheat, especially when combined with mineral fertilization. At the same time, the obtained results concerning the content of pigments and proline indicated that mineral fertilization primarily increased nutrient availability, while the introduction of bacteria likely functioned through other mechanisms, possibly related to stress resistance.

### 3.3. Mechanisms of Action of Enterobacter ludwigii Strain GMG_278 on Wheat

The results of the previous experiments led us to the concept that the effect of the *Enterobacter ludwigii* GMG_278 strain is most likely not related to an increase in nutrient availability. For this reason, we decided to analyze some key points (Figure 5) of the response of plants inoculated with the *Enterobacter ludwigii* GMG_278 strain to stress factors (drought or salinity stress) under conditions of mineral supplementation or without mineral fertilizers.

The PCA method made it possible to identify three groups of genes that responded differently to the introduction of bacterial fertilizer. The first group showed virtually no change in gene expression. This group corresponded to the initial stages of abiotic stress perception and included genes of ABA receptors and ethylene-sensitive factors. Phytohormones such as salicylic acid (SA), jasmonic acid (JA), ethylene (ET), and ABA are endogenous molecules with low molecular weight, which primarily regulate plant defense reactions against both biotic (predominantly ET, SA, and JA) and abiotic (predominantly ABA) stress [41]. ABA accumulates rapidly in response to various environmental stress conditions, and ABA-deficient plants have altered stress responses. ABA promotes stomatal closure, inhibits stomatal opening to decrease water loss through transpiration, and induces the expression of numerous genes related to the accumulation of proteinogenic amino acids and some sugars [42]. Under limited water supply, plants increase the production of stress-generating ethylene (ET) through the active disintegration of a metabolic precursor. Ethylene is an inhibitor of plant growth; in particular, it inhibits seed germination and root development [43].

Additionally, the first group included the MAPK and WKY26 genes, which are part of the signal transduction pathways from stress phytohormone receptors. In all eukaryotes, mitogen-activated protein kinase (MAP) (MAPK/MPK) cascades are highly conserved central regulators of various cellular processes, such as differentiation, proliferation, growth, death, and stress responses. In plants, the MAPK cascade plays a crucial role in various biotic and abiotic stress responses, as well as in hormonal responses that include ROS signaling [44] and proline accumulation [45]. Ultimately, we can conclude that the effect of the *Enterobacter ludwigii* GMG_278 strain was not associated with changes in ethylene or ABA production, despite the fact that the accomplished effect was similar to that provided by these pathways.

The second group of genes exhibited minor changes in expression in response to the introduction of the strain. This group included the transcription factors DREB and WKY71; their modulation caused changes in the levels of stress-related metabolites. The effects of these transcription factors are mainly associated with the production of phytohormones such as IAA, gibberellins, ethylene, ABA, and cytokinins. These phytohormones, in addition to playing an important role in plant growth and development, can help plants cope with abiotic stresses. We observed an increase in the expression of auxin-responsive factor and cytokinin dehydrogenase genes. Auxin mitigates drought stress indirectly by increasing root growth and/or modifying the architecture of roots and/or root hairs, which facilitates the uptake of water and nutrients from the soil [43]. Drought typically causes a decrease in cytokinin levels that has at least two major consequences, namely increased sensitivity to ABA and reduced shoot growth [46].

The third group of genes showed a very strong increase in their expression in response to the introduction of bacteria. These were three genes involved in ROS defense and related to enzymes POD, CAT, and LPX. Rapid ROS production plays a key role in both ABA signaling and biotic stress resistance responses. Moreover, it is considered that ROS detoxification through the action of enzymes such as superoxide dismutase, catalase, and glutathione peroxidase, as well as non-enzymatic antioxidant components such as ascorbic acid and glutathione, neutralize the cytotoxic effects of ROS in various stress situations [47].

A more detailed assessment of the stressful impacts on plants revealed a number of features under studied experimental conditions. In particular, attention should be paid to the genes of the first group, which were activated in response to drought stress in accordance with their functional significance. At the same time, the genes of the third group were activated to a lesser extent under drought effects than under salinity stress or stress-free conditions. This fact attested to a smaller amount of ROS, which was probably produced to a lesser extent due to some other mechanisms of stress resistance against drought.

## 4. Materials and Methods

### 4.1. Isolation, Screening, and Identification

Soil samples were assembled for the civil scientific project “Atlas of Soil Microorganisms of Russia” from three different areas: Krasnodar (GMG_378), Sakhalin (GMG_278, GMG_294), and Novosibirsk (GMG_336). Soil microbes were isolated from each sample by the spread plate method. Then, 0.1 g of each soil sample was dispersed in 50 mL of sterilized NaCl (0.9%) and thoroughly shaken using a vortex mixer. Afterward, 100 mkl was spread onto corresponding agar plates.

NFB isolation was carried out on Ashby agar plates with 2% glucose and a trace amount of bromothymol blue (BTB) at 30 ± 1 °C. After 3 to 10 days of incubation, colonies with changes in the color of the medium were recorded.

PSB isolation was carried out on Pikovskaya’s agar medium (PVK) containing insoluble tricalcium phosphate and a trace amount of bromophenol blue at 30 ± 1 °C. After 3 to 10 days of incubation, colonies with changes in the color of the medium were recorded.

SPB isolation was carried out on CAS-agar plates at 30 ± 1 °C for 48 h. The siderophore-producing bacterial colonies showed an orange color around the colony. The CAS assay is based on a siderophore’s ability to bind to ferric iron with high affinity. An indication of siderophore production was the color changing from blue to purple (as described in the traditional CAS assay for siderophores of the catechol type) or from blue to orange (as reported for microorganisms that produce hydroxamates) in halo shapes around the colonies [48].

Different colonies were chosen and purified using the subculture method on the respective agar media to obtain pure colonies. Colony morphology and color were recorded after 24 h of growth. Bacterial identification was initially performed using Gram staining reactions, then examined using a microscope.

Next, the molecular identification of the isolates was determined on the basis of 16S rDNA sequence analysis. Bacterial isolates were cultured for 48 h, and the DNA of the isolates was extracted according to the procedure described by Sambrook et al. [49]. The DNA template for PCR amplification was prepared by picking the individual colony of each strain and amplification of 16S rRNA gene. Amplification of the gene was carried out by PCR using 27F (5-AGAGTTTGATCTTGGCTCAG-3) and 1492R (5-GGT TAC CTT GTT ACG ACT T-3). The purification of PCR products and sequencing were carried out at the SB RAS Genomics Core Facility (http://www.niboch.nsc.ru/doku.php/sequest, accessed on 1 September 2024). The 16S rDNA gene sequences of the bacterial isolates obtained were matched with available gene sequences using BLAST (http://www.ncbi.nlm.nih.gov, accessed on 4 September 2024) and aligned by employing the Clustal W program. Phylogenetic trees were constructed using the neighbor-joining method, and molecular evolutionary analyses were conducted using the MEGA X software (version 10.2.5) [50].

### 4.2. Quantitative Assessment of Potential Properties That Promote Plant Growth

For each experiment, a microorganism was cultured for 24 h at 28 °C in triplicate, using an appropriate growth medium. A medium without inoculum was used as a control. A peptone-based medium was employed to measure the production of gibberellins, ammonium, and siderophores, as well as proline and salicylic acid. Additionally, a peptone medium supplemented with L-tryptophan at a concentration of 50 mg/100 mL was used to assess auxin production. To measure phosphate solubilization, a Pikovskaya medium was utilized. Cells were centrifuged at 10,000× *g* for 10 min, after which the supernatant was collected and used for analysis. To evaluate tolerance to salt and heavy metals, a bacterial suspension was employed. After the addition of appropriate reagents, the wavelength was measured using a spectrophotometer (Varioskan Flash, Thermo Fisher Scientific, St. Louis, MO, USA). An average value was determined from three replicates.

#### 4.2.1. Solubilization of Insoluble Phosphate

P-solubilization was quantified via the phospho-molybdate blue color method using a spectrophotometer (λ = 882), as described by Murphy and Riley [51]. For quantitative evaluation, a comparison with the standard curve obtained using a standard solution of potassium phosphate was used.

#### 4.2.2. Production of Ammonia

Nesseler’s reagent (10 μL) was added to 200 μL of the culture supernatant. The development of a brown-to-yellow color indicated ammonia production. The absorbance was measured at 450 nm. For quantitative assessment, a standard curve generated using a standard ammonium sulfate solution was employed.

#### 4.2.3. Production of Indole-3-Acetic Acid

Bacterial isolates were inoculated in sterilized nutrient broth supplemented with 1% tryptophan (precursor for IAA production), then incubated in a shaker for 3 days at 28–30 °C. After the incubation period, the cultures were centrifuged at 10,000 rpm for 10 min before 1 mL of each supernatant was mixed with 2 mL Salkowski reagent (1 mL of 0.5 M FeCl_3_ in 50 mL of 35% HClO_4_) [52]. The mixtures were left at room temperature for 30 min. The development of a pink color indicated the production of IAA, and the quantification of IAA was read at 530 nm. A standard curve was plotted for the quantification of the IAA solution and uninoculated medium, with a reagent serving as a control.

#### 4.2.4. Siderophore Production

Culture supernatant and a Chromeazurol S, at a ratio of 1:1, were used as references. The percentage of siderophore units produced was calculated using the following formula: % siderophore unit = [(Ar − As)/Ar] × 100, where Ar = absorbance of the reference at 630 nm and As = absorbance of the sample at 630 nm [53].

#### 4.2.5. Production of Proline

A mixture of acidic ninhydrin reagent (each ml of which contained 0.4 mL of 6 M orthophosphoric acid, 0.6 mL of glacial acetic acid, and 25 mg of ninhydrin), glacial acetic acid, and a sample of culture supernatant in equal proportions was incubated in a water bath at 100 °C for 1 h. The optical density was measured at a wavelength of 520 nm. For quantitative evaluation, a comparison was made with a standard curve obtained by a similar method using standard proline solutions [54].

#### 4.2.6. Production of Salicylic Acid

The synthesis of SA in broth culture can be determined by the previously described method [55]. The culture medium was adjusted to a pH of 2.0 by adding 1N HCl, and then SA was extracted from the culture supernatant using chloroform. The ratio of culture supernatant to chloroform was 1:2, and the extraction was carried out by vigorous shaking. For quantitative analysis, 200 μL of distilled water, 200 μL of 2M FeCl_3_, and 200 μL of the extracted chloroform fraction were combined. This resulted in the formation of a violet Fe-SA complex in the aqueous phase. The absorption of this complex was then measured at a wavelength of 527 nm, using SA dissolved in the growth medium as a reference.

#### 4.2.7. Production of Gibberellins

Gibberellic acid was determined based on the method described by Abou-Aly et al. [56] as follows: 100 μL of 30% hydrochloric acid (HCl) and 100 μL Folin–Ciocalteau reagent were added to 100 μL culture supernatant in a clear test tube. Then, 300 μL distilled water was added, and the mixture was heated at 100 °C for 5 min in a thermostat. After cooling, the intensity of the resulting bluish-green color was measured using a spectrophotometer at 760 nm. A standard solution of GA3 was also prepared, and its color was measured in the same manner.

#### 4.2.8. Biofilm Formation

The determination of biofilm formation was carried out on the basis of the previously described method [57]. A volume of 200 μL of a bacterial suspension in peptone media was transferred to each well of a plate and incubated for 24 h at 37 °C. Following this, the contents of each well were removed and washed twice with PBS, followed by drying. The wells were then stained with aqueous gentian violet (0.1%) solution for 30 min. After staining, the gentian violet solution was carefully removed, and 200 μL of 96% ethanol solution was added and pipetted into each well. The optical density of the solution was then measured at 570 nm using a spectrophotometer.

### 4.3. Whole Genome Sequencing

#### 4.3.1. WGS Methodology

The genomic library from the isolate DNA sample was prepared using the NEBNext Ultra II DNA Library Prep Kit for Illumina (NEB) reagents with some modifications of the manufacturer’s protocol. 1000 ng of genomic DNA was fragmented to 400–500 bp on a Covaris S220 device in 100 µL of sterile water. The fragmented DNA was purified and concentrated using AMPure XP (Beckman Coulter, USA) magnetic particles, the particles were mixed with DNA in a ratio of 1.6 to 1, respectively. Purified and fragmented DNA was used in the reactions of end completion, adenylation of 3′ ends, and ligation of the NEBNext Adapter for Illumina (NEB) adaptors. DNA with sewn adapters was processed by USER Enzume (NEB) to remove uracil from the adapter and “open” the hairpin structure, which was then purified using AMPure XP (Beckman Coulter) magnetic particles. The particles were mixed with DNA in a ratio of 0.9 to 1, respectively. Then, amplification (3–6 PCR cycles) of the resulting library was performed, during which the adapter sequences were completed and index sequences were included in them. The qualitative assessment of the obtained libraries was carried out on the Agilent TapeStation 4150 bioanalyzer using the High Sensitivity D5000 ScreenTape and High Sensitivity D5000 Reagents (Agilent, USA) kits, quantitative—using real-time PCR using the KAPA Library Quantification Kit (KAPA Biosystems, USA) reagent kit. The resulting genomic library was sequenced on the Illumina NovaSeq 6000 device in the mode of pair-terminal readings 151 + 151 bp.

#### 4.3.2. Assembly De Novo

The quality of the sequencing was assessed using the FastQC v.0.12.1 software (https://www.bioinformatics.babraham.ac.uk/projects/fastqc/ (accessed on 12 June 2024)). Low-qualities reads were filtered (Q > 28, minlength > 100), and Illumina adapters were trimmed by Trimmomatic v.0.39 [58]. Genome assembly was performed for pair-end reads using the SPAdes v3.15.5 software using parameters -k 21, 33, 55, 77, and -o isolate according to the recommendations of the developers [59]. The resulting contigs were manually filtered. The filtration threshold for the average coverage level was assessed empirically. First, all the contigs with a coating of less than 5 and a length of less than 300 were filtered, and then the coverage of the remaining ones was manually evaluated. The threshold was considered to be the level of coverage equal to the minimum coverage of extended contigs (70×). The species was determined by the blastn tool on the 16S_ribosomal_RNA database (ftp://ftp.ncbi.nlm.nih.gov/blast (accessed on 12 June 2024)).

#### 4.3.3. Quality Control and Species Verification

First, the species was determined by the blastn tool on the 16S_ribosomal_RNA database (ftp://ftp.ncbi.nlm.nih.gov/blast (accessed on 23 September 2023)). The best score result was used for the next two rounds of verification of taxonomy. The second step was the analysis using OrthoANI [60]. To compare the de novo genome assembly, sequences of complete genomes with the taxonomy the same as the 16S best score analysis were downloaded from the NCBI database. The strains were considered to belong to the same species when the sequences matched by more than 97%.

#### 4.3.4. Genome Annotation

The prediction of the open reading frames (ORFs) in the de novo assembly was carried out using GeneMark.hmm software version 3.38 [61]. The resulting list of ORFs was analyzed by the software BlastKoala (https://www.kegg.jp/blastkoala/ (accessed on 18 August 2024)) [62] and InteProScan (https://interproscan-docs.readthedocs.io/en/latest/UserDocs.html (accessed on 18 August 2024)) [63]. Since the fullness of different databases varies, it is advisable to search through all databases in order to identify genes whose protein products are annotated only in some databases. With the help of InterProScan, the assembly was annotated according to the following databases: AntiFam-7.0, CDD-3.20, Coils-2.2.1, FunFam-4.3.0, Gene3D-4.3.0, Hamap-2023_05, MobiDBLite-2.0, NCBIfam-14.0, PANTHER-18.0, Pfam-37.0, PIRSF-3.10, PIRSR-2023_05, PRINTS-42.0, ProSitePatterns-2023_05, ProSiteProfiles-2023_05, SFLD-4, SMART-9.0, and SUPERFAMILY-1.75.

The identification of genes that encode proteins with known domains but differ significantly in nucleotide sequence from homologous genes was performed using the following algorithm. First, the list of open reading frames generated by GeneMark.hmm software (https://genemark.bme.gatech.edu/license_download.cgi (accessed on 18 August 2024)) was aligned to the nt_prok database using the blastn tool. The open reading frames that could not be aligned were analyzed by the program InterProScan (https://interproscan-docs.readthedocs.io/en/latest/UserDocs.html (accessed on 18 August 2024)). The result of the analysis was a list of domains identified in these proteins.

### 4.4. Plant Inoculation and Experimental Design of Pot Trial

The effect of bacterial inoculation on plant growth was studied on “Novosibirsk 31” wheat varieties in a pot experiment under greenhouse conditions. Pots with diameters of 10 cm that could hold 0,25 kg of soil were used in this experiment. The soils were sterilized at a temperature of 121 °C (103 kPa) for 30 min, hermetically incubated at a temperature of 22 °C for 24 h, and autoclaving was repeated.

Five seeds were placed in each pot at a depth of 2–3 cm. All the selected seeds were surface-sterilized with 1% NaOCl for 90 s and two consecutive rinses in sterile distilled water, followed by air-drying under laminar air flow.

Bacterial cultures were grown in 50 mL falcon tubes filled with 10 mL LB broth and were kept in a shaker at 200 rpm for 48 h, then diluted to adjust 108 cfu/mL bacterial solutions with sterile, distilled water. Seeds were coated with culture by immersion in a suspension of bacteria for 120 min. This experiment was carried out with three replications, and the results were compared with control seeds treated with water instead of a bacterial isolate.

The control pots (Water) were irrigated with water. In the case of adding mineral fertilizer (MN), watering was performed with the following solution: N-108, P-39, K-117, Ca-120, Mg-28, and S-36,4 (with the addition of chelates of elements Fe, Mn, Zn, Cu, Mo, and B). For the stress experimental groups, irrigation was carried out using the following solutions: to simulate drought—12% PEG 6000 (PEG) and to simulate salinization—1,2% NaCl (NaCl). In the case of the simultaneous use of mineral fertilizer and stress (MN-PEG and MN-NaCl), the PEG and NaCl solutions were prepared using mineral solution. Irrigation with PEG and NaCl solutions was started two weeks after planting. In all cases, watering was performed as soon as the soil dried out. The experiment was set up as a randomized design with three biological replications.

### 4.5. Measurements of Plant Parameters

The plant growth lasted for 30 days. During the growing period, the average temperature was 22 °C, and the relative humidity fluctuated between 70% and 80%. After the experiment was completed, agronomic parameters were measured, including plant height (cm), root and aboveground biomass (g), and dried root and aboveground biomass (g).

The content of chlorophyll a and b, carotenoids was determined as described in [64] (extraction method in 95% ethanol), and the proline content was determined as described in [65] (Colorimetric Assay).

The total RNA of the wheat roots was extracted from different samples using a kit for RNA isolation and purification from plants R-PLANTS (BioLabmix, Novosibirsk, Russia). The concentration and purity of the RNA were verified by measuring the absorbance at 260/280 nm. Reverse transcription and real-time polymerase chain reaction (RT-RT-PCR) were performed in a one-step method using a kit «BioMaster RT-qPCR SYBR Blue» (BioLabmix, Novosibirsk, Russia). The list of primers, their sources, and their roles in plant metabolism are presented in Supplementary Matherials (Appendix A). A thermal cycler CFX C1000 (BioRad, Hercules, CA, USA) using a cycling program of 95 °C for 180 s, 45 cycles of 95 °C for 5 s, 60 °C for 5 s, and 20 s at 72 °C. The expression of genes under study was computed by using the threshold (Ct) value for each gene normalized against the Ct for actin from wheat, which was used as the constitutive reference transcript. The relative expression levels of all samples were calculated and analyzed based on the 2^−∆∆CT^ method [66].

### 4.6. Statistical Analysis

Comparing the groups for statistical differences in these data, the significance was tested using ordinary one-way ANOVA analysis and Tukey’s multiple comparisons test by GraphPad Prism 10 (https://www.graphpad.com/features, accessed on 12 August 2024). Venn diagram was built using the R package VennDiagram [67].

A principal component analysis (PCA) was performed to analyze the relationships between soil isolates and parameters measured by tests. PCA was performed using the R procomp() function with standard parameters (https://www.rdocumentation.org/packages/stats/versions/3.6.2/topics/prcomp, accessed on 12 May 2023). The isolates from the soil were grouped into clusters on a two-dimensional graph of two main components: PCA1 and PCA2.

A clustered heat map was built using the pheatmap v. 1.0.12 function in the R package [68].

## 5. Conclusions

The analysis of the properties of four strains of *Enterobacter ludwigii* bacteria that positively affected plants showed a significant diversity of traits. These discrepancies were not consistent with the analysis of the whole genome, where the revealed differences mainly concerned the synthesis of carbohydrate components and protection against viruses. Meanwhile, the assessment of their effects on plants in the pot experiment revealed the most promising strain, GMG_278. This suggests that genome analysis does not always accurately reflect the functional properties of bacteria or their mechanisms of action on plants. This is due to the insufficient annotation of many bacterial genes and the limited understanding of the mechanisms driving their positive effects; therefore, when analyzing genomic data, it is crucial not only to consider the genetic information but also to focus on phenotypic traits.

The investigation of wheat gene expression during inoculation with the *Enterobacter ludwigii* GMG_278 strain led to the activation of antioxidant protection in wheat, as well as to the rearrangement of the synthesis of some phytohormones to increase root growth or modify the architecture of root hairs. In turn, additional mineral fertilizing presented a significant increase in the biomass of the above-ground part of plants. This result should also have a positive effect on plant resistance to phytopathogens and stress factors, as we have seen in the example of drought and salinity stress.

It should be noted that the strain GMG_278 showed relatively low indices on ammonium production and phosphate solubilization in phenotypic tests; consequently, the properties enhancing plant nutrition were not very pronounced in this strain. That has been confirmed by experiments assessing the content of different pigments; however, this strain showed good production of gibberellins, which explained the substantial increase in plant biomass. In addition, GMG_278 showed the highest production of salicylic acid that attested to the significant activation of antioxidant defense in plants in response to inoculation with this strain.

Therefore, future research should focus on analyzing oxidative stress activation pathways as a key factor in selecting plant-beneficial bacteria.

## Figures and Tables

**Figure 1 plants-13-03551-f001:**
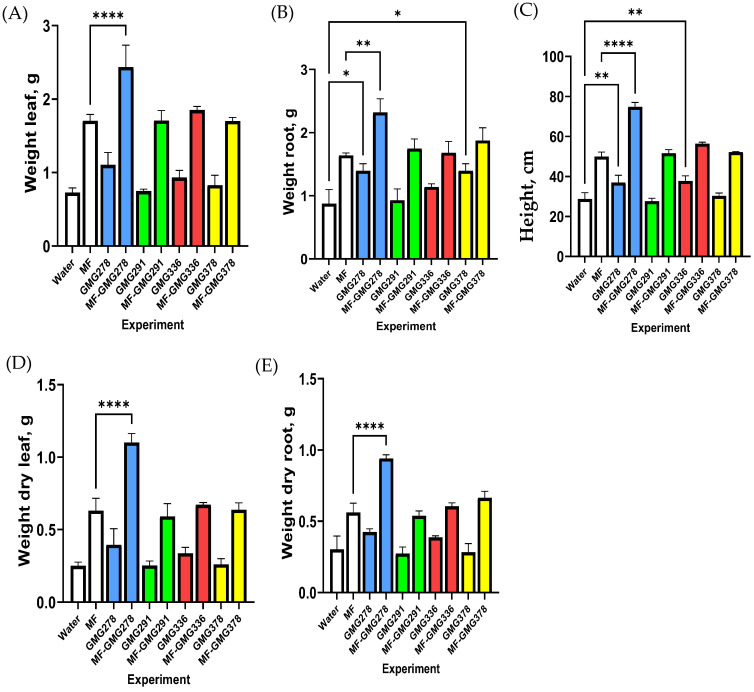
Diagrams of agronomic indicators of plants after treatment with bacteria and mineral fertilizer. (**A**) the weight of the aboveground part of the plant, (**B**) the weight of the underground part of the plant, (**C**) the height of the plant, (**D**) the dry weight of the aboveground part of the plant, (**E**) the dry weight of the underground part of the plant. *—*p* (*p* values) < 0.05, **—*p* < 0.01, ****—*p* < 0.0001.

**Figure 2 plants-13-03551-f002:**
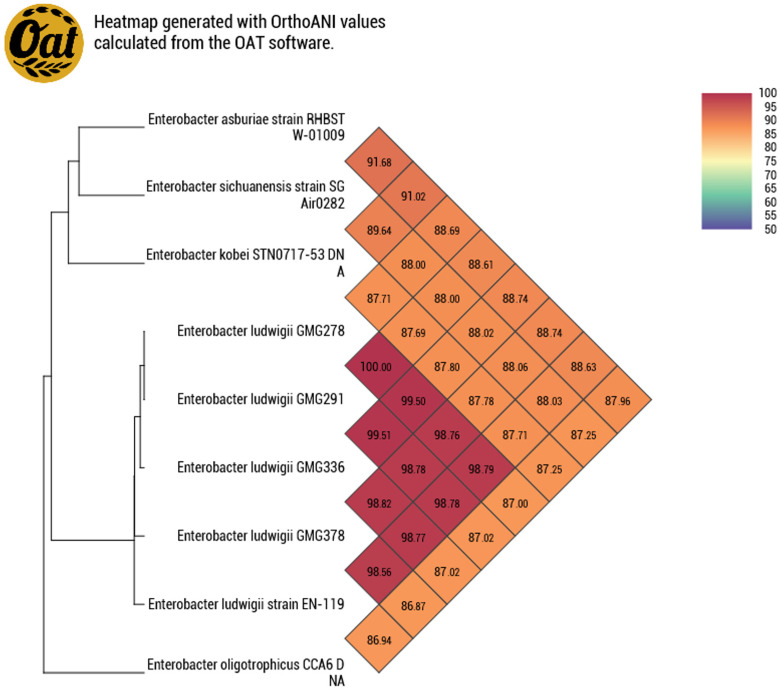
Alignment to the complete genomes of *Enterobacter* sp. by OrtoANI analysis. OrthoANI measures the overall similarity between two genome sequences. The threshold for species discrimination is 95~96%. The genomes for comparison were downloaded from the online accessible database https://www.ncbi.nlm.nih.gov/datasets/genome/ (accessed on 6 October 2024).

**Figure 3 plants-13-03551-f003:**
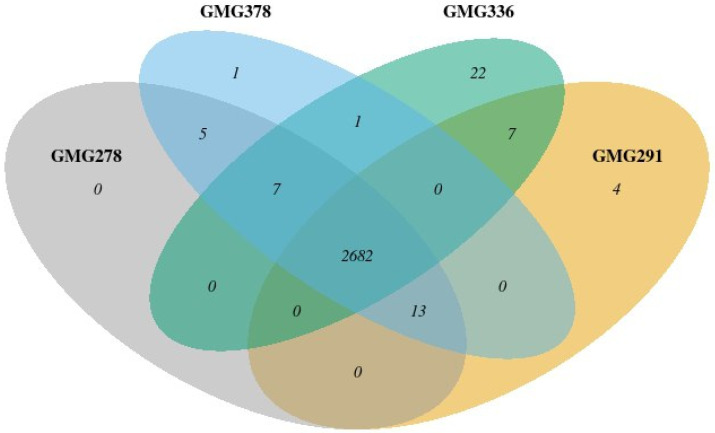
The Venn diagram shows the number of common genes in the studied strains.

**Figure 4 plants-13-03551-f004:**
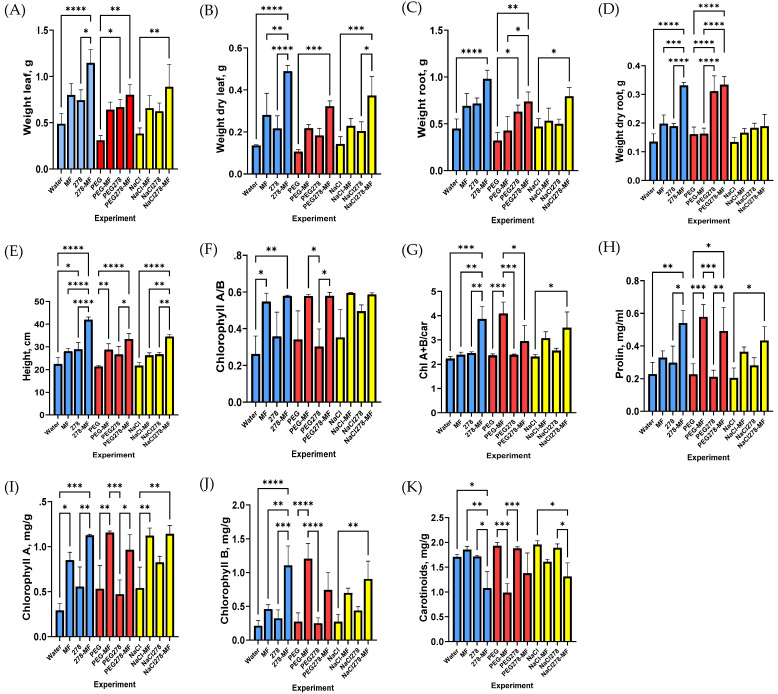
Diagrams of morphological and physiological parameters of plants under stress conditions in experiments. Legend (MF—mineral fertilizer, 278—microbial fertilizer, NaCl—watering with 1% NaCl solution, PEG—watering with 10% PEG solution). (**A**,**B**) the weight of the wet (**A**) and dry (**B**) aboveground part of the plant; (**C**,**D**) the wet weight (**C**) and dry weight (**D**) of the underground part of the plant; (**E**) height of the plant; (**I**–**K**) the amount of pigment: chlorophyll A (**I**), chlorophyll B (**J**), carotenoids (**K**); (**H**) the amount of proline; (**F**,**G**) quantity ratio ChlA/ChlB (**F**), ChlA+B/car (**G**). *—*p* (*p* values) < 0.05, **—*p* < 0.01, ***—*p* < 0.001, ****—*p* < 0.0001.

**Figure 5 plants-13-03551-f005:**
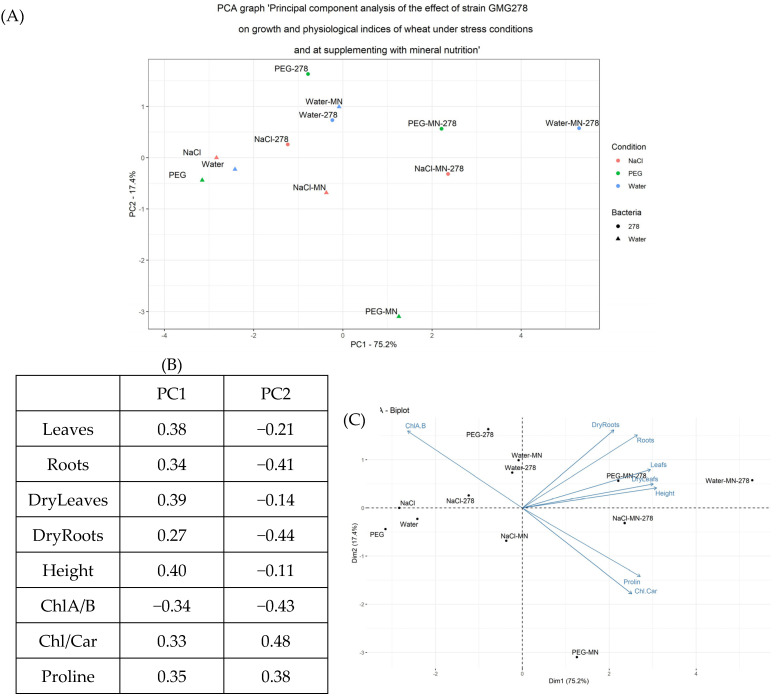
Principal component analysis of the effect of strain GMG_278 on growth and physiological indices of wheat under stress conditions and at supplementing with mineral nutrition. Legend (MN—mineral nutrition, 278—microbial fertilizer, NaCl—watering with 1% NaCl solution, PEG—watering with 10% PEG solution). (**A**) PCA plot, (**B**) the contribution of indicators to the main component, (**C**) Redundancy analysis (RDA).

**Figure 6 plants-13-03551-f006:**
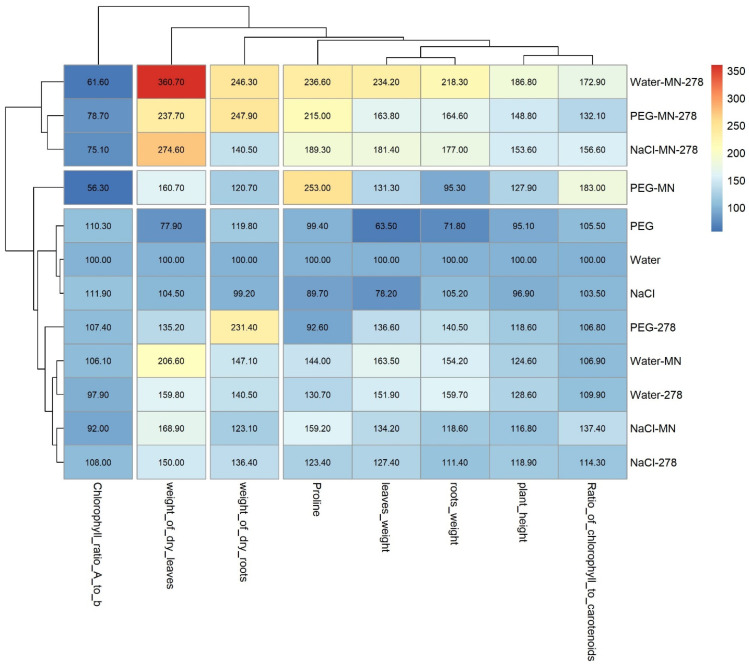
Heat map of changes in the morphological and physiological characteristics of wheat grown under stress conditions and at supplementing plants with mineral nutrition and the GMG_278 strain. Legend (MN—mineral nutrition, 278—microbial fertilizer, NaCl—watering with 1% NaCl solution, PEG—watering with 10% PEG solution).

**Figure 7 plants-13-03551-f007:**
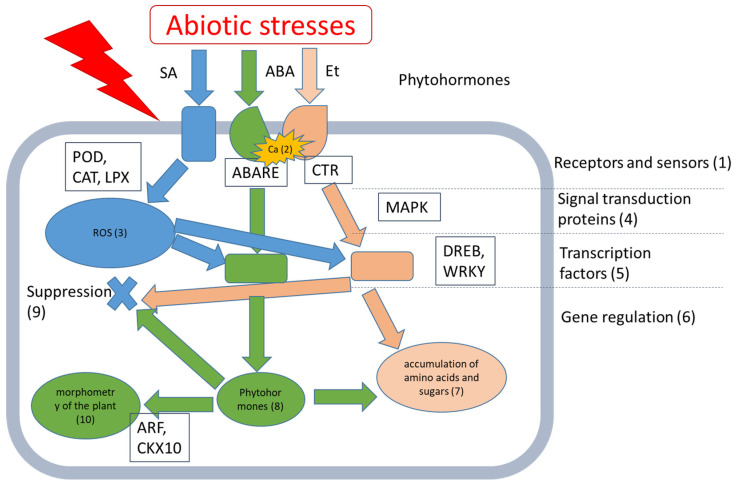
Schematic representation of the effects of stress factors on plants. Most of the mechanisms for sensing and initiating adaptive responses to various abiotic stresses involve changes in proteins and lipids in biological membranes (1). Unfavorable conditions entail ultrastructural changes in biomolecules, which are sensed by receptors or specialized proteins, leading to the augmentation of the Ca^2+^ level in the cytosol (2), as well as REDOX imbalance (3). These signals activate kinase cascades and other secondary events that stimulate phosphorylation/dephosphorylation consecutive events (4), reaching a culmination in TF activation (5), and remodeling of gene expression (6). Moreover, the activation of certain enzymes stimulates the biosynthesis of osmolytes, pigments, thermoprotectants (7), and other secondary metabolites (8) in addition to ROS detoxifying enzymes (9), which are aimed to restore redox homeostasis in stressed cells. At the systemic level, various transformations are shown. They concern morphological changes, which are detected in the root and leaves and affect the proliferation of lateral roots in response to drought; biochemical changes associated with the secretion of phytochelants in response to heavy metals; and also the reclusion and closure of stomata in addition to a decrease in leaf area and abscission (10). The figure is based on the previous study [21].

**Figure 8 plants-13-03551-f008:**
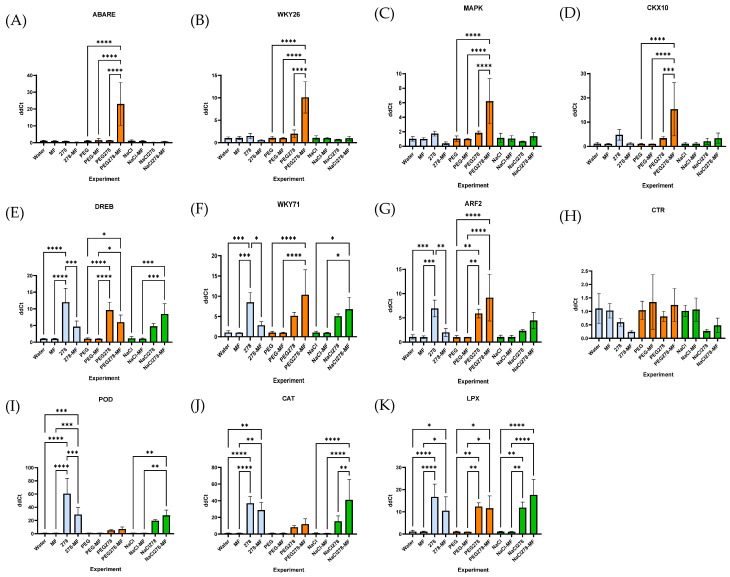
Diagrams of the dependence of gene expression involved in response to stress factors in wheat grown under drought and salinity conditions on introducing GMG_278 strain. Legend (MF—mineral fertilizer, NaCl—watering with 1% NaCl solution, PEG—watering with 10% PEG solution). (**A**) gene ABARE, (**B**) gene WKY26, (**C**) gene MAPK, (**D**) gene CKX10, (**E**) gene DREB, (**F**) gene WKY71, (**G**) gene ARF2, (**H**) gene CTR, (**I**) gene POD, (**J**) gene CAT, and (**K**) gene LPX. *—*p* (*p* values) < 0.05, **—*p* < 0.01, ***—*p* < 0.001, ****—*p* < 0.0001.

**Figure 9 plants-13-03551-f009:**
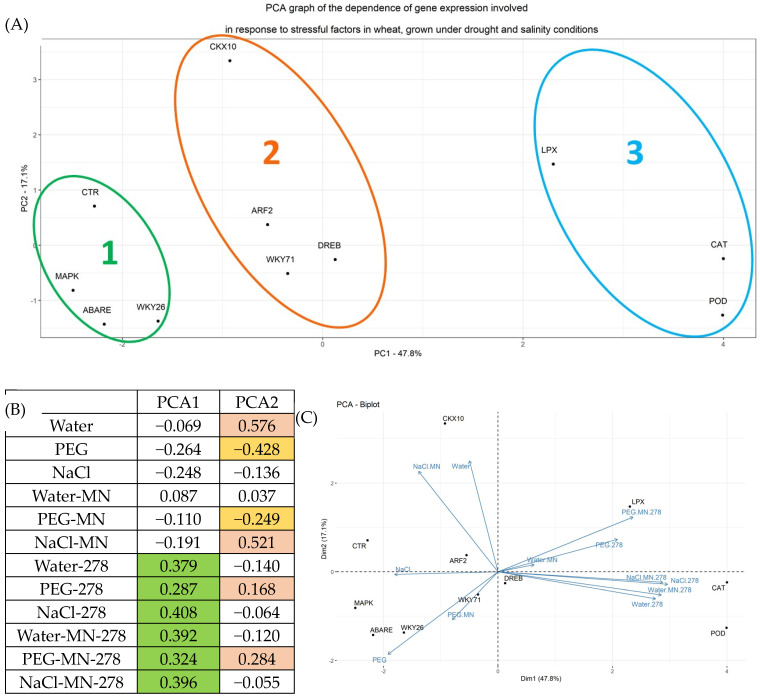
Principal component analysis of data on the expression of genes involved in response to stress factors in wheat grown under drought and salinity conditions at introducing GMG_278 strains. Legend (MN—mineral nutrition, 278—microbial fertilizer, NaCl—watering with 1% NaCl solution, PEG—watering with 10% PEG solution). (A) PCA plot, (B) the contribution of indicators to the main component. The color indicates the maximum absolute coefficients for each PSA vector. For PSA1—green, for PSA 2 two colors are used: pink for conditions that shift the gene expression level up along the PSA2 axis, yellow—down along the PSA2 axis. (C) Redundancy analysis (RDA).

**Table 1 plants-13-03551-t001:** PGP features of the studied strains of the *Enterobacter ludwigii* species (maximum values among the studied strains are bolded).

Strain	Phosphate Solubilization, µg/mL	Ammonia Production, µg/mL	Siderophore Production, %	IAA Production, µg/mL	GA Production, µg/mL	Proline Production, µg/mL	SA Production, µg/mL	Biofilm Formation, OD570
GMG278	53.97 ± 16.58	9.46 ± 2.81	15.83 ± 3.54	47.56 ± 8.77	**3.19 ± 0.57**	14.17 ± 2.95	**1.19 ± 0.45**	0.4 ± 0.04
GMG291	106.82 ± 22.64	**19.49 ± 1.83**	15.37 ± 3.48	59.4 ± 6.54	2.49 ± 0.36	**21.72 ± 5.86**	0.16 ± 0.04	**0.42 ± 0.05**
GMG336	**112.08 ± 12.64**	17.21 ± 4.92	17.09 ± 5.77	**73.19 ± 13.51**	2.31 ± 0.34	**21.11 ± 9.19**	0.14 ± 0.01	0.28 ± 0.06
GMG378	53.66 ± 13.45	16.99 ± 2.72	**19.53 ± 5.09**	69.27 ± 3.04	2.47 ± 0.29	16.5 ± 8.81	0.13 ± 0.01	0.22 ± 0.04

**Table 2 plants-13-03551-t002:** General characteristics of the genomes of *Enterobacter ludwigii* strains.

Strain Code	GMG_278	GMG_291	GMG_336	GMG_378
Strain Name	*Enterobacter ludwigii* strain AF137-NN-B1	*Enterobacter ludwigii* strain AF137-PP-C2	*Enterobacter ludwigii* strain AF-SC-144.2.1	*Enterobacter ludwigii* strain AF-SC-P-D6.1
GenBank ACCESSION	JBHGBZ000000000	JBHGCA000000000	JBHGCB000000000.1	JBHGCC000000000.1
BioProject	PRJNA1146973	PRJNA1146973	PRJNA1146973	PRJNA1146973
BioSample	SAMN43141185	SAMN43141414	SAMN43153427	SAMN43154427
Total base	4,750,581	4,750,417	4,688,440	4,833,514
GC (%)	54.74	54.74	54.77	54.33
Coverage, min	70×	70×	50×	40×
Genes (total)	4535	4520	4450	4674
CDSs (total)	4452	4449	4367	4595
Genes (coding)	4408	4406	4328	4530
CDSs with protein	4408	4406	4328	4530
Genes (RNA)	83	71	83	79
rRNAs	1, 1 (5S, 16S)	1, 1 (5S, 16S)	1, 1 (5S, 16S)	2, 1 (5S, 16S)
complete rRNAs	1, 1 (5S, 16S)	1, 1 (5S, 16S)	1, 1 (5S, 16S)	2, 1 (5S, 16S)
tRNAs	71	61	71	70
ncRNAs	10	8	9	6
Pseudo genes	44	43	39	65
CDS (without protein)	44	43	39	65
Pseudo Genes (ambiguous residues)	0 of 44	0 of 43	0 of 39	0 of 65
Pseudo Genes (frameshifted)	22 of 44	22 of 43	18 of 39	25 of 65
Pseudo Genes (incomplete)	28 of 44	27 of 43	27 of 39	49 of 65
Pseudo Genes (internal stop)	11 of 44	11 of 43	9 of 39	16 of 65
Pseudo Genes (multiple problems)	12 of 44	12 of 43	11 of 39	21 of 65

**Table 3 plants-13-03551-t003:** List of genes annotated in the KeggOrthology database, which distinguish strains GMG_378, GMG_336, and GMG_291 from strain GMG_278.

Strain	KEGG Orthology	KEGG Genes	KEGG Pathway	The Area of Work of Genes
GMG_291	K24302	UDP-N-acetylbacillosamine transaminase	O-Antigen nucleotide sugar biosynthesis	glycopeptide and glycolipid synthesis genes
	K12996	rhamnosyltransferase	Lipopolysaccharide biosynthesis proteins O-antigen repeat unit	glycopeptide and glycolipid synthesis genes
	K10906	exodeoxyribonuclease VIII	DNA repair and recombination proteins	repair genes
	K01160	crossover junction endodeoxyribonuclease RusA	DNA repair and recombination proteins	repair genes
GMG_336	K00879	ATP:L-fuculose 1-phosphotransferase	Carbohydrate metabolism	carbohydrate metabolism genes
	K01818	L-fucose/D-arabinose isomerase	Carbohydrate metabolism	carbohydrate metabolism genes
	K01628	L-fuculose-phosphate aldolase	Carbohydrate metabolism	carbohydrate metabolism genes
	K00048	lactaldehyde reductase	Carbohydrate metabolism	carbohydrate metabolism genes
	K02431	L-fucose mutarotase	Racemases and epimerases	glycopeptide and glycolipid synthesis genes
	K13002	glycosyltransferase	Lipopolysaccharide biosynthesis proteins O-antigen repeat unit	glycopeptide and glycolipid synthesis genes
	K18799	O-antigen flippase	Lipopolysaccharide biosynthesis proteins O-antigen repeat unit	glycopeptide and glycolipid synthesis genes
	K03277	heptosyltransferase IV	Lipopolysaccharide biosynthesis proteins	glycopeptide and glycolipid synthesis genes
	K06952	5′-nucleotidase	Nucleotide metabolism	repair genes
	K00986	RNA-directed DNA polymerase	DNA polymerases	repair genes
	K06919	putative DNA primase/helicase	Replication and repair	repair genes
	K02429	MFS transporter, FHS family, L-fucose permease	Transporters	efflux pump genes
	K13408	membrane fusion protein	Plant-pathogen interaction	efflux pump genes
	K13409	ABC transporters	Plant-pathogen interaction	efflux pump genes
	K18842	antitoxin ChpS	Toxin-antitoxin system (TA system)	efflux pump genes
	K01322	prolyl oligopeptidase	Serine endopeptidases	
	K25227	linear primary-alkylsulfatase	Sulfuric-ester hydrolases	
	K06946	uncharacterized protein		
	K09960	uncharacterized protein		
	K07474	phage terminase small subunit		
	K14744	prophage endopeptidase	Viral proteins	
	K06988	8-hydroxy-5-deazaflavin:NADPH oxidoreductase	Acting on the CH-NH group of donors	
GMG_378	K06938	uncharacterized protein		

**Table 4 plants-13-03551-t004:** Genes unique to the studied strains.

Strain	Database	ID	Domain Name	Metabolic pathways	The Area of Work of Genes
GMG_278	Pfam	PF01177	Asp/Glu/Hydantoin racemase	biosynthesis of peptidoglycan and some peptide-based antibiotics such as gramicidin	Glycopeptide and glycolipid synthesis genes
	Pfam	PF14552	Tautomerase enzyme	ket-enol tautomerization reaction, carbon, and energy	carbohydrate metabolism genes
	Pfam	PF07693	KAP family P-loop domain	One possible function of the prokaryotic KAP NTPases might be in the exclusion of selfish replicons, such as viruses, from the host cells [14].	antiviral defense genes
	Pfam	PF13289	SIR2-like domain	is a component of the bacterial antiphage defense system Thoeris and has robust NAD+ cleavage activity [15]	antiviral defense genes
	Pfam	PF03466	LysR substrate binding domain	The structure of this domain is known and is similar to the periplasmic binding proteins [16]	genes for communication with the external space
	SUPERFAMILY	SSF52540	P-loop containing nucleoside triphosphate hydrolases	P-loop motif is common to many ATP and GTP-binding proteins and is similar in structure to that found within adenylate kinase [14]	genes for communication with the external space
GMG_291	Pfam	PF14552	Tautomerase enzyme		carbohydrate metabolism genes
	Pfam	PF01177	Asp/Glu/Hydantoin racemase		Glycopeptide and glycolipid synthesis genes
	Pfam	PF07693	KAP family P-loop domain		antiviral defense genes
	Pfam	PF13289	SIR2-like domain		antiviral defense genes
	Pfam	PF03466	LysR substrate binding domain		genes for communication with the external space
GMG_336	Pfam	PF14552	Tautomerase enzyme		carbohydrate metabolism genes
	Pfam	PF01177	Asp/Glu/Hydantoin racemase		Glycopeptide and glycolipid synthesis genes
	Pfam	PF03466	LysR substrate binding domain		genes for communication with the external space
GMG_378	ProSiteProfiles	PS51257	Prokaryotic membrane lipoprotein lipid attachment site profile	Synthesis of lipoproteins	Glycopeptide and glycolipid synthesis genes
	Pfam	PF00535	Glycosyl transferase family	The biosynthesis of disaccharides, oligosaccharides and polysaccharides	Glycopeptide and glycolipid synthesis genes
	Pfam	PF00534	Glycosyl transferases group 1	The biosynthesis of disaccharides, oligosaccharides and polysaccharides	Glycopeptide and glycolipid synthesis genes
	Gene3D	G3DSA:3.40.50.2000	Glycogen Phosphorylase B	glucosyltransferase family	Glycopeptide and glycolipid synthesis genes
	Pfam	PF01943	Polysaccharide biosynthesis protein	production of polysaccharide	Glycopeptide and glycolipid synthesis genes
	PANTHER	PTHR30471	DNA REPAIR PROTEIN RADC		repair genes
	Pfam	PF13392	HNH endonuclease	Bacterial HNH domains are known to act as both site-specific DNA nucleases (in homing nucleases, for example, or in the RNA-guided DNA endonuclease Cas9) as well as non-specific nucleases (in colicins, a subgroup of bacterial toxins) [17]	antiviral defense genes
	Pfam	PF19040	SGNH domain (fused to AT3 domains)	resistance to viruses and antimicrobials, and biosynthesis of antibiotics [18]	antiviral defense genes

## Data Availability

Data are contained within the article and Appendix A.

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
