# Peer review of "Phenotypic and Genomic Analysis of *Enterobacter ludwigii* Strains: Insights into Mechanisms Enhancing Plant Growth Both Under Normal Conditions and in Response to Supplementation with Mineral Fertilizers and Exposure to Stress Factors"

_plants, 2024, doi:10.3390/plants13243551_

Round 1

Reviewer 1 Report

Comments and Suggestions for Authors

You have shown how much work you have to do when looking for a future PGPB! Congratulations to you and your research team. I enjoyed reading your article and was surprised at how much work you put into it. The data you presented is insightful and I hope you will continue the good work in the future. I have one suggestion regarding Figures 9a and 9b. There is a lack of clarity that can be improved.

Author Response

Reply to Reviewer #1

You have shown how much work you have to do when looking for a future PGPB! Congratulations to you and your research team. I enjoyed reading your article and was surprised at how much work you put into it. The data you presented is insightful and I hope you will continue the good work in the future. I have one suggestion regarding Figures 9a and 9b. There is a lack of clarity that can be improved.

Dear Reviewer,

The co-authors and I would like to thank you for your appreciation of our research work. This will give us enthusiasm for future research!

We have clarified the description of Figures 9a and 9b in the updated version of the manuscript.

With best wishes,

Ms. Voronina E.N.

Reviewer 2 Report

Comments and Suggestions for Authors

The manuscript relates a useful study on plant growth promoting bacteria but this version of its presentation detracts from the work in ways that can be corrected. Some suggestions follow:

1. Rewording for clarity and grammar changes throughout the text to improve readability would be advisable, especially in the Introduction and Materials and Methods sections where communication of the objectives and specifics of the work are critical.

2. Other grammatical changes include rewriting run-on sentences (e.g. line 32 of the introduction) and sentences beginning with a number (e.g. line 606 of section 4.2.8). Also, line 492 of section 3.3 appears to be written in Russian.

3. Formatting of references (e.g. references 5-7 of the introduction section) requires global review and modification.

4. Text citations of Figures and Tables are either missing or incorrectly formatted (e.g. Figure 9 is not cited in the text before the figure appears and Table 4 citation may be missing also).

5. Species names should be in italics throughout.

6. Figure 3 could be left off perhaps--its purpose is unclear but in any case, it needs clarification in the figure legend.

7. Figure 7 should be redesigned or discarded.

8. Table 5 might better go in the Supplemental.

Comments on the Quality of English Language

As described above, grammatical changes and formatting corrections should render the manuscript more readable.

Author Response

Reply to Reviewer #2

Dear reviewer,

The co-authors and I would like to thank you for your careful reading of the manuscript and for very helpful and instructive comments. We have made appropriate revisions to the manuscript and hope that it will be suitable for the next round of review. We have included the responses point by point below.

The manuscript relates a useful study on plant growth promoting bacteria but this version of its presentation detracts from the work in ways that can be corrected. Some suggestions follow:

  1. Rewording for clarity and grammar changes throughout the text to improve readability would be advisable, especially in the Introduction and Materials and Methods sections where communication of the objectives and specifics of the work are critical.

We have corrected the wording in the Introduction and Materials and Methods sections. The corrections are highlighted in yellow.

  1. Other grammatical changes include rewriting run-on sentences (e.g. line 32 of the introduction) and sentences beginning with a number (e.g. line 606 of section 4.2.8). Also, line 492 of section 3.3 appears to be written in Russian.

We agree; in the new version of the manuscript, we have made edits in the indicated places, highlighted in yellow.

The paragraph is indeed written in Russian, probably there was a technical failure when forming the file. It has been removed from the new version of the manuscript.

  1. Formatting of references (e.g. references 5-7 of the introduction section) requires global review and modification.

The reference list was carefully reviewed and reformatted according to the journal's requirements.

  1. Text citations of Figures and Tables are either missing or incorrectly formatted (e.g. Figure 9 is not cited in the text before the figure appears and Table 4 citation may be missing also).

We have carefully checked the references to figures and tables and made changes.

  1. Species names should be in italics throughout.

We have corrected this defect throughout the text.

  1. Figure 3 could be left off perhaps--its purpose is unclear but in any case, it needs clarification in the figure legend.

We clarified the figure legend.

  1. Figure 7 should be redesigned or discarded.

This figure shows a selection of genes that were examined for changes in expression in response to bacterial inoculation, mineral fertilization, drought and salinity stress. We agree that it is not completely accurate, but we have tried to reflect the key points of involvement of these genes in the development of wheat response to external influences. It seems that for people who do not have sufficient information about the function of these genes, it will be useful to see the diagram in front of them to better understand the functional significance of these genes. Especially, since we are moving Table 5 to the supplementary material.

  1. Table 5 might better go in the Supplemental.

The comment has been taken into account and the table has been moved to Supplemental.

With best regards,

Mrs. Voronina EN

Reviewer 3 Report

Comments and Suggestions for Authors

In order to comprehend how Enterobacter ludwigii strains can promote plant growth under both normal and stressful environmental conditions, the authors examined the phenotypic and genomic traits of four of these bacteria in this study. Phosphate solubilization, ammonium production, iron phosphate production, and phytohormone synthesis are examples of plant growth promoting (PGP) traits that strains were chosen for based on their capacity to grow in nitrogen-free media.

Despite performing a genomic analysis, the strains' genomic diversity did not show any appreciable variations in known genes linked to advantageous plant growth traits. This is intriguing because it raises the possibility that the mechanisms of action are more intricate or involve genes that have not yet been identified.
The absence of gene annotations was mentioned by the authors as a potential drawback. To provide a more comprehensive understanding of how these strains promote plant growth, it may be helpful to talk about alternate functional mechanisms or think about looking into the expression of particular genes further.

Also, disparities between the phenotypic and genomic data were noted in the study, which is a common problem in microbial research. For instance, although the strains exhibited some phenotypic activity in nitrogen-fixing tests, genomic data did not fully support nitrogen fixation.
To find out why the phenotypic tests don't always match the genomic predictions - particularly when it comes to nitrogen metabolism - this could be investigated further.

Furthermore, although the authors shed light on the function of salicylic acid and other phytohormones, the paper would benefit from a more thorough examination of the genetic underpinnings of the bacteria's capacity to stimulate plant growth.
The manuscript might also benefit from a stronger link between the physical processes that result in improvements in plant growth and the observed changes in gene expression.

More thorough tables or visual aids, like Venn diagrams or heatmaps, would help some sections - especially those that deal with genomic analysis and phytohormone production - by making it easier to compare data from different strains.

If possible, please add a few details:

- There are not enough references in the introduction.
- Throughout the manuscript, the names of the species should be italicized.
- Strain is not the first row in Table 1, but it is the first column.
Remaking the table would be preferable.
- 2.1 Enterobacter ludwigii strain selection and phenotypic screening which seems to belong in the experimental section more.
- All things considered, the figures' quality needs to be raised, and each one lacks detailed explanations.

Comments on the Quality of English Language

An English editing service is needed

Author Response

Dear reviewer,

We are very grateful to you for your careful reading of our article and your interest in the results obtained. We fully agree that some of the data can and should be discussed and studied in more depth. And in the process of writing, an attempt was made more than once to divide the article into several parts in order to have the opportunity for a more detailed discussion. However, we wanted to present the work on studying these strains in a "complete" form, from the initial screening to the description of the complete genome and studies of the mechanism of action on plants. It seems to us that it is in this vein that the problems arising when working with microorganisms that have a positive effect on plants are maximally emphasized. In the future, we plan to conduct more in-depth studies of individual areas and issues outlined in this article. The amount of illustrative material was also reduced as much as possible due to its large volume.

If possible, please add a few details:

- There are not enough references in the introduction.

We have added links in the introduction to support the statements made.

- Throughout the manuscript, the names of the species should be italicized.

Done

- Strain is not the first row in Table 1, but it is the first column. Remaking the table would be preferable.

Done

- 2.1 Enterobacter ludwigii strain selection and phenotypic screening which seems to belong in the experimental section more.

This section presents the results of studies of the degree of manifestation of traits useful for plants. Quite significant differences in these traits of these strains is the result of the study, which is further discussed in the light of the full genome sequencing data. Since we did not initially select strains by these traits, but took all the strains of Enterobacter ludwigii from the collection, this section is still more suitable for the "Results" section.

- All things considered, the figures' quality needs to be raised, and each one lacks detailed explanations.

We have improved the quality of the drawings. For Figures 2, 3 and 9, the legend has been expanded.

With best regards,

Mrs. Voronina EN

Round 2

Reviewer 3 Report

Comments and Suggestions for Authors

The species name or genus name must be written in italics, but terms like Enterobacter or Zea mays or Fusarium are still not written in this way. 

In expressions like L-fucose/D-arabinose isomerase, the 'L' and 'D' should be written as small capital letters.

Comments on the Quality of English Language

Some sentences are complex in their structures. It would be beneficial to revise them for improved clarity and readability.
